# A Conventional Cruise and Felled-Tree Validation of Individual Tree Diameter, Height and Volume Derived from Airborne Laser Scanning Data of a Loblolly Pine (*P. taeda*) Stand in Eastern Texas

Mark V. Corrao [1,*], Aaron M. Sparks [1] and Alistair M. S. Smith [1,2]

1   Department of Forest, Rangeland, and Fire Sciences, College of Natural Resources, University of Idaho, Moscow, ID 83844, USA; asparks@uidaho.edu (A.M.S.); alistair@uidaho.edu (A.M.S.S.)
2   Department of Department of Earth and Spatial Sciences, College of Science, University of Idaho, Moscow, ID 83844, USA
*   Correspondence: mcorrao@uidaho.edu

**Abstract:** Globally, remotely sensed data and, in particular, Airborne Laser Scanning (ALS), are being assessed by the forestry industry for their ability to acquire accurate forest inventories at an individual-tree level. This pilot study compares an inventory derived using the ForestView® biometrics analysis system to traditional cruise measurements and felled tree measurements for 139 *Pinus taeda* sp. (loblolly pine) trees in eastern Texas. The Individual Tree Detection (ITD) accuracy of ForestView® was 97.1%. In terms of tree height accuracy, ForestView® results had an overall lower mean bias and RMSE than the traditional cruise techniques when both datasets were compared to the felled tree data (LiDAR: mean bias = 1.1 cm, RMSE = 41.2 cm; Cruise: mean bias = 13.8 cm, RMSE = 57.5 cm). No significant difference in mean tree height was observed between the felled tree, cruise, and LiDAR measurements (*p*-value = 0.58). ForestView-derived DBH exhibited a −2.1 cm bias compared to felled-tree measurements. This study demonstrates the utility of this newly emerging ITD software as an approach to characterize forest structure on similar coniferous forests landscapes.

**Keywords:** LiDAR; DBH; ITD; felled tree; loblolly; field data

## 1. Introduction

Accurate forest inventories are needed to forecast growth and yield over large spatial scales, especially in areas where on-the-ground field data is lacking due to limited access or high acquisition cost. To meet this need, airborne scanning Light Detection and Ranging (LiDAR) scanning, also referred to as Airborne Laser Scanning (ALS), is widely used to accurately acquire forest inventories at the stand [1] and individual-tree level within the commercial forestry sector [2–5]. Numerous Individual Tree Detection (ITD) algorithms exist; however, few have been applied at a landscape scale [1,3,6,7], in part due to a lack of high-resolution ALS data (e.g., 20 pulses per square meter (ppm)) and costly field validation data. A further challenge is that robust validation of ITD algorithm-derived tree diameters is limited but essential, given the use of these metrics in growth and yield models [8]. Within the commercial forest industry, quantification of tree-stem volume is of significant physiological and financial importance for growth and yield modeling and sustainable forest management [9–11].

In the southern United states, *Pinus taeda* L. (*P. taeda*), commonly known as loblolly pine, is of high economic importance due to its fast growth rates and relatively high stem volume [12–14]. Common approaches for modeling tree stem volume rely on allometric equations that utilize easy-to-measure attributes like tree height and diameter at breast height (DBH) [15]. Allometric equations are typically non-linear, thus errors in tree height and DBH measurements can have significant effects on the accuracy of stem volume

calculations [1,16]. Unquantified errors in tree height and DBH can negatively impact the accuracy of modeled tree volume [16,17], above ground biomass [18,19], and site index calculations [20], all of which are core elements in the management and valuation of commercial forests.

Technological developments in LiDAR sensors and analytical techniques have led to many recent advances in ITD-derived forest inventories. However, despite these advances, independent accuracy assessments of these algorithms remain needed to give end-users information on the suitability and accuracy of the algorithm products. Prior research has demonstrated the high accuracy of ITD algorithms for individual tree height estimates at plot [6] and stand scales [1,5,21]. Several studies have also presented accuracy assessments of ITD-derived heights [8,22] to indirect field measured heights from traditional inventory methods [23,24], and direct measured heights such as those from felled tree research [25,26]. Specifically, several studies have shown ALS-derived tree heights are typically more accurate and less biased than field-measured tree height [24,27].

In previous research, ALS-derived tree height measurements and accuracies have varied depending on LiDAR sensor, ALS flight specifications, and forest structure [28,29]. These studies, and others (i.e., [3]), further demonstrated ITD-derived attribute accuracy is influenced by the LiDAR sensor platform, data density, and analytical methods, further highlighting the need to quantify the accuracies within emerging LiDAR-based ITD software packages. Accurate tree height measurements, and DBH estimates [30–32] from ALS data are of significant importance to the forest industry's application of ITD methods for silvicultural and forestland investment decisions [5,33]. Thus, while previous studies of ALS-based forest metrics have been compared to field measurements and felled tree data (e.g., [3,7,25,29,34,35]), no studies have assessed the ForestView® ITD software's ability to accurately measure and map height, DBH, and volume within commercial *P. taeda* plantations.

This pilot study compares the results of an emerging and proprietary ALS-based ITD software, ForestView®, to field measurements within a mature stand of *P. taeda* in eastern Texas. ForestView®, developed by Northwest Management Incorporated (NMI, Moscow, Idaho), which is described in [2], is an individual tree detection and attribute measurement software that has been applied on >1.1 million hectares of commercial forestland within the United States and Canada. In their research [2] only assessed ForestView® on a mixed conifer forest in northern Idaho, limiting the ability to evaluate whether the method is broadly applicable across the diversity of United States commercial forest types. Therefore, the objectives of this study are to extend that initial analysis and:

(1) Assess the ability of ForestView® to provide comparable or improved height, DBH and volume measurements at the individual tree level in a *P. taeda* stand;

(2) Provide an estimate of total gross volume by tree for forest valuation and merchandizing considerations.

## 2. Materials and Methods

### 2.1. Study Area

The study area was a 0.5-hectare even-aged *P. taeda* plantation in Cherokee County, approximately 44 km northwest of Nacogdoches, Texas, Latitude 31.75° N Longitude −95.09°W (Figure 1). The measurement stand was comprised of six rows planted at a 3.7 m (~12 feet) 3 m square spacing. A private landowner, participating in a multi-state 600,000-hectare ALS-based forest inventory, offered access to measure the remaining trees (139) being harvested from the target stand. Elevation of this stand was 210 m above sea level, the soil composition was a well-drained upland Betis loam [36] and the topography exhibited a slight north to northeast-facing aspect. The subject stand was established in 1999 as a single-species plantation and was commercially thinned to remove every 5th row prior to field measurement. The stand had a site index of 67 (base 25 year). Data collection occurred over the course of two days due to heavy rains and inaccessible road conditions

as well as the duration of time needed to collect quality field data for every tree, complete UAV flights, and maintain a safe distance from the mechanized harvester as it cut each tree.

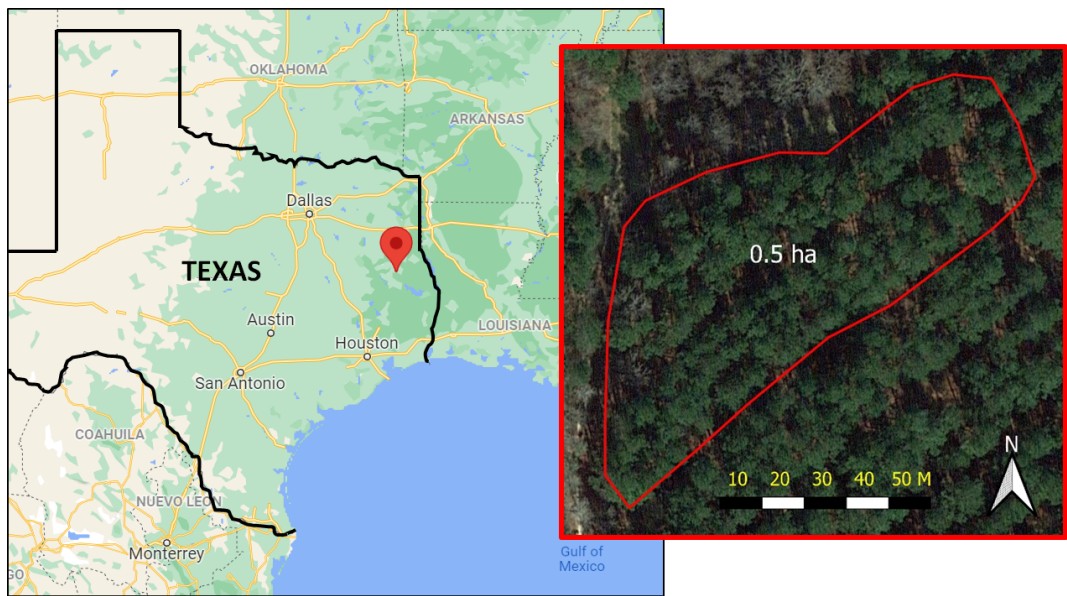

**Figure 1.** Study area ~44 km northwest of Nacogdoches, TX where LiDAR, field cruise, UAV imagery, and felled tree data were collected.

### 2.2. ALS Data and Preprocessing

ALS data were acquired in December 2020 using a RIEGL VQ-1560II sensor (RIEGL, Horn, Austria) mounted on a fixed-wing aircraft fitted with a Gyro Stabilization 4000 Mount (SOMAG, Jena, Germany). Elevation of the aircraft was maintained between 1600 and 1900 m above ground level and flight lines alternated orientations while maintaining a 50% flight-line overlap with respect to the 58-degree sensor field-of-view. The average scan density was 20 pulses per square meter with an average pulse return rate of four over forested landscapes. The ALS data were preprocessed to normalize laser intensity within the RIEGL RiPROCESS software (RIEGL, Horn, Austria) and classified to bare earth, vegetation, water, buildings, and noise returns before being tiled into 500 m$^2$ .LAZ file-type tiles by the LiDAR acquisition company and delivered to the landowner, and available for our team.

### 2.3. ALS Individual Tree Detection and Measurement

The ALS .LAZ files were imported into ForestView® for individual tree detection and processing of stand- and individual tree-metrics. Individual tree detection within this software begins with generation of a digital elevation model (DEM) and digital surface model (DSM) in order to generate a canopy height model (CHM) at 0.5 m resolution directly from the ALS point cloud. The software then iterates through multiple methods, similar to watershed and local maxima algorithms [21,37,38], that detect peaks in the CHM. These peaks are assumed to be the tops of tree "approximate" objects thus their location and respective height are recorded. For tree attribute estimation, the software relies on an internal database of field- and ALS-measured stem mapped trees, each having DBH, height, species, crown condition, and taper information. The software calculates a large number (~100+) of metrics from the ALS point cloud for each individual tree object, and uses these metrics along with the field measured attributes in the database to model tree attributes for each ALS-detected tree. The DBH modeling draws on height-, crown-, density-, and spacing-related metrics derived from the ALS point cloud data and their respective allometric relationships to the trees stem mapped in the field. Further details on ForestView® processing and outputs are reported in [2]. Individual tree location, maximum

height, and DBH were the only ForestView®-derived Digital Inventory® metrics utilized in this research. Additional outputs available but not assessed in this study were various crown descriptors, height to live crown, social dominance, and gross stem volume. Canopy cover for the study area was calculated as the number of ALS first returns above 1.37 m, divided by the total number of first returns.

*2.4. Field Validation Dataset*

Field measurements included a standing-tree inventory cruise of every tree prior to felling and measurements taken after each tree was felled. Measurements were acquired in May 2021 generally following the approach of [32] for height, DBH, and social position. Tree height, height-to-live-crown from the soil surface, azimuth from plot center, and average canopy width to the nearest 0.3 m were acquired using a Vertex Laser Geo 360 hypsometer (HAGLOF, Langsele, Sweden). A logger's tape was used to measure DBH to the nearest 0.25 cm, and a Spiegel relaskop (SILVANUS, Kirchdorf, Austria) was used to gather a bole height, where 80% of DBH was used to approximate taper following the methods of [39]. Additionally, the location of each tree was further recorded using a JAVAD Triumph-2 (JAVAD EMS, Silicon Valley, CA, USA) with sub-meter precision and differentially corrected in postprocessing. Each tree was numbered sequentially and painted with high-visibility tree marking paint at DBH and at the base. Video footage of individual trees within the study area was acquired using an unmanned aerial vehicle (UAV), Phantom 4 Pro (DJI, Shenzhen, China), equipped with a 4K camera (DJI, Shenzhen, China). The UAV was flown at various heights and flight patterns (e.g., 30 m above the canopy and within the canopy to capture individual trees from each cardinal direction at varying distances). The resulting imagery served as another tree-matching data source.

After measurement and GPS-recording each tree's location, every tree was felled with a mechanical harvester and laid near its respective stump. All heights were remeasured with a logger's tape to the nearest 1 foot (0.3 m). An upper-stem diameter, accounting for stump height, was measured at 32 feet (9.75 m) to 1/10th inch (~0.25 cm) outside-of-bark with a steel logger's tape and calipers. All trees measured in the field where manually matched to the corresponding ForestView® detected tree coordinates using a combination of the JAVAD location, a 1-foot (0.3-m) resolution canopy height model, notes taken by the cruiser, and the UAV video footage captured prior to felling.

Lastly, given the temporal discrepancy between the ALS data collection (December 2020) and the felled tree data (May 2021), height and diameter growth from December 2020 to May 2021 were modeled for each tree. Height and DBH growth modeling was informed using *P. taeda* growth data from near Nacogdoches, TX [12] and the Forest Projection System (FPS) software *P. taeda* species model [40]. These inputs to FPS included all individually measured trees within the study area as a representative sample plot. This allowed the model, to generate estimated increases in height and DBH. The adjusted height and DBH values were then applied to each tree and used throughout the following comparative analyses. Individual tree volume was calculated by the FPS software referencing the tree species, height and DBH metrics gathered in the field.

*2.5. Height, DBH and Volume Accuracy Assessment*

Multiple statistical tests were applied to evaluate the statistical similarity of ALS-derived and forester-cruised height and DBH to felled tree height and DBH. A Kolmogorov–Smirnov test was used to assess whether the ALS-derived height and DBH distributions were statistically similar to the felled tree height and DBH distributions. The Kolmogorov–Smirnov test identifies differences in the cumulative distribution function of two data distributions, where the test statistic (D) symbolizes the point of maximum discrepancy in the cumulative distribution functions. D values closer to 0 indicate significant overlap in the data distributions and values closer to 1 indicate little to no overlap in the data distributions.

Regression-based equivalence tests [41] were used to assess the accuracy of ALS-derived height and DBH. Regression-based equivalence tests evaluate regression slope and intercept equality between paired sets of data. Intercept equality implies that the mean of one dataset is not significantly different than the mean of another dataset. Slope equality refers to the similarity of pairwise measurements, if these measurements are equal, the regression will be characterized by a slope of 1. Following similar studies [2,41,42], the region of equality was set to ±25% of the mean for the intercept and ±25% for the slope. The null hypothesis of dissimilarity is rejected if the region of equivalence contains two joint one-sided 95% confidence intervals for the intercept or slope. Regression intercept and slope equality were assessed for each of the paired height and DBH datasets (i.e., ALS-derived vs. felled data, cruised vs. felled data). Additional accuracy measures, including average root mean square error (*RMSE*) (Equation (1)) and *Mean Bias* (Equation (2)) were calculated between ALS-derived and forester-cruised and felled tree measurements as follows:

$$RMSE = \sqrt{\frac{\sum_{i=1}^{n}(\hat{x}_i - x_i)^2}{n}} \tag{1}$$

$$Mean\ Bias = \frac{1}{n}\sum_{i=1}^{n}(\hat{x}_i - x_i) \tag{2}$$

where $(\hat{x}_i)$ are the predicted values, $x_i$ are the observed values, and $n$ is the number of observations. All statistical analyses were conducted in R [43], and the 'equivalence' R package [44] was used to conduct the regression-based equivalence tests.

## 3. Results

The results of the manual tree matching are provided in (Table 1) and displayed in (Figure 2a,b). The study area had an average canopy cover of ~86% and ForestView® identified 153 individual trees (Figure 2a) compared to the 139 GPS-located *P. taeda* (Figure 2b). During field measurements some non-conifer stumps were observed. The forestry crew preparing the stand for measurement reported the removal of multiple Liquidambar styraciflua (Sweet gum) trees and number of large mid-story shrubs prior to the research team's arrival. These trees and shrubs were not tracked in the field for alignment with the ALS-derived Digital Inventory.

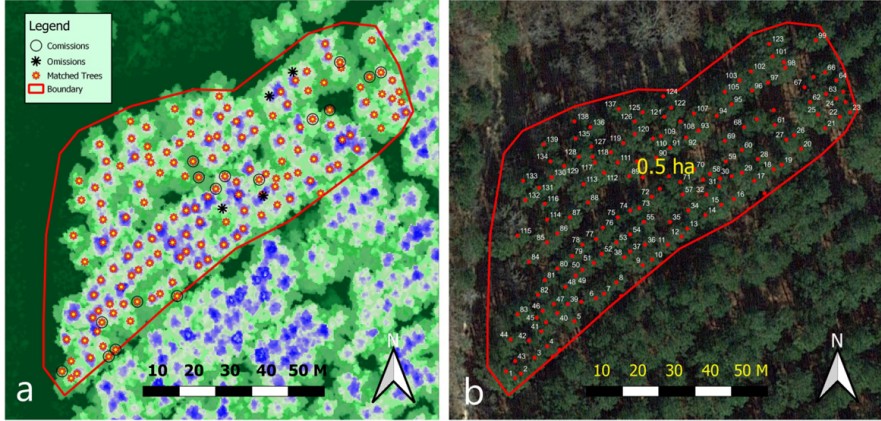

**Figure 2.** The resulting spatial alignment of the ALS 0.3 m canopy height model and the cruised and felled tree data. (**a**) There were 153 trees identified by ForestView®, three trees were not detected (omission error), and 15 extra trees were identified by ForestView® (commission error). ALS commission errors were assumed to be attributable to a mix of deciduous tree species and mid-canopy shrubs mechanically removed from the site prior to field measurements. (**b**) There were 139 loblolly pine (*P. taeda*) trees field-mapped (cruised) and mechanically felled within the study area. The sequentially numbered individual tree labels are provided for reference.

**Table 1.** Summary of December 2020 ALS, May 2021 cruise data, and May 2021 felled tree measurement data within the 0.5 ha study site ~44 km northwest of Nacogdoches, Texas (Latitude 31.75 Longitude −95.09).

| Attribute | LiDAR | Cruise | Felled Tree |
|---|---|---|---|
| Total Detected Trees | 153 | 139 | 139 |
| LiDAR Omissions | 3 | - | - |
| LiDAR Commissions | 15 | - | - |
| No. of Matched Felled Trees | 135 | 139 | 139 |
| Detection Rate (%) | 97.1 | 100 | 100 |
| Mean Height (m) [a] | 21.0 | 21.1 | 20.9 |
| Min Height (m) [a] | 17.2 | 17.4 | 17.1 |
| Max Height (m) [a] | 23.3 | 23.8 | 23.2 |
| SD Height (m) [a] | 1.2 | 1.3 | 1.2 |
| Mean DBH (cm) [b] | 33.2 | 32.8 | - [c] |
| Min DBH (cm) [b] | 22.9 | 19.8 | - [c] |
| Max DBH (cm) [b] | 40.6 | 45.5 | - [c] |
| SD DBH (cm) [b] | 3.7 | 4.6 | - [c] |
| Mean Gross Volume (m$^3$) | 0.64 | 0.63 | 0.63 |
| Min Gross Volume (m$^3$) | 0.24 | 0.19 | 0.19 |
| Max Gross Volume (m$^3$) | 0.98 | 1.09 | 1.06 |
| SD Gross Volume (m$^3$) | 0.15 | 0.18 | 0.19 |
| Total Gross Volume (m$^3$) | 85.89 | 85.28 | 84.88 |

[a] ITD heights derived from LiDAR plus growth increment from December to May. [b] ITD DBH derived from LiDAR plus growth increment from December to May. [c] DBH values for felled trees were represented by cruise measurements of standing trees for consistency purposes.

The average height of ITD trees from ForestView® and in-field felled tree measurements were within 0.01 m and exhibited similar distributions (Table 1, Figure 3).

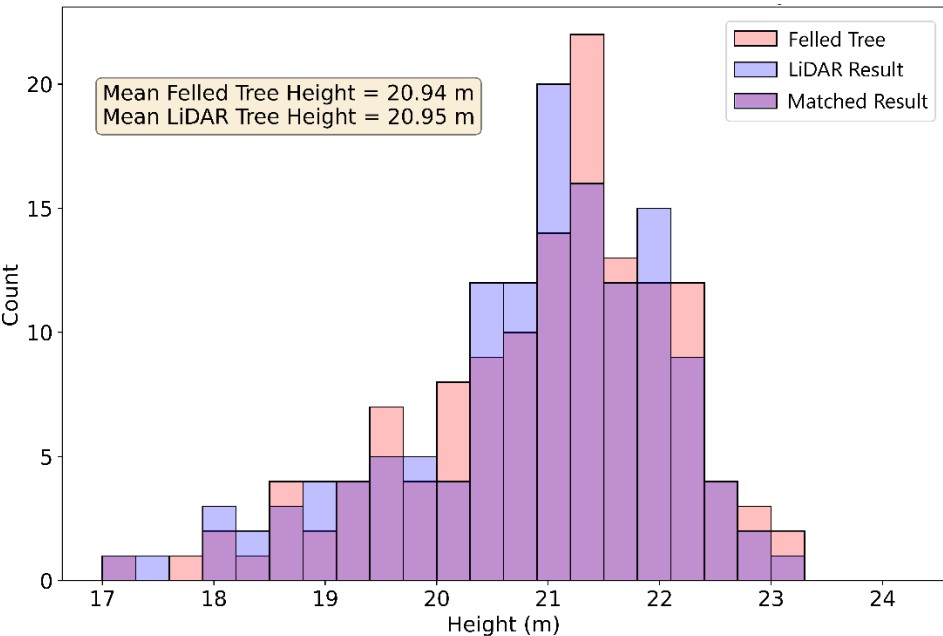

**Figure 3.** ForestView® growth-adjusted tree height distribution (blue) compared to felled-tree field measured height distribution (red). The (purple) color is where these two datasets overlap, representing a "matched result".

The ALS-derived height of detected trees exhibited a −0.34 m bias and RMSE of 0.53 compared to the felled tree measurement data, whereas the ALS-derived growth-

adjusted height exhibited a 0.01 m bias and RMSE of 0.41 m compared to the felled tree measurement data. The cruised tree height exhibited a 0.14 m bias and RMSE of 0.58 m compared to the felled tree measurement data. For both paired tree height datasets (ALS-derived vs. felled and cruised vs. felled), the 95% confidence interval for the intercept (vertical red bar) was within the $\pm 25\%$ region of equivalence (grey polygon), indicating that the null hypothesis of dissimilarity can be rejected (i.e., paired datasets are equivalent). Likewise, for both paired datasets, the 95% confidence interval for the slope (vertical black bar) was within the $\pm 25\%$ region of equivalence (grey dashed lines), indicating slopes were significantly similar to 1. The linear relationship between ALS-derived and felled tree height (Figure 4a) had $R^2$ value of 0.89, while the relationship between ALS-derived growth-adjusted height and felled tree height (Figure 4b) had a $R^2$ value of 0.81. The relationship between cruised and felled tree height (Figure 4c) had a $R^2$ value of 0.81. Figure 5 presents results from the regression-based equivalence tests for tree height.

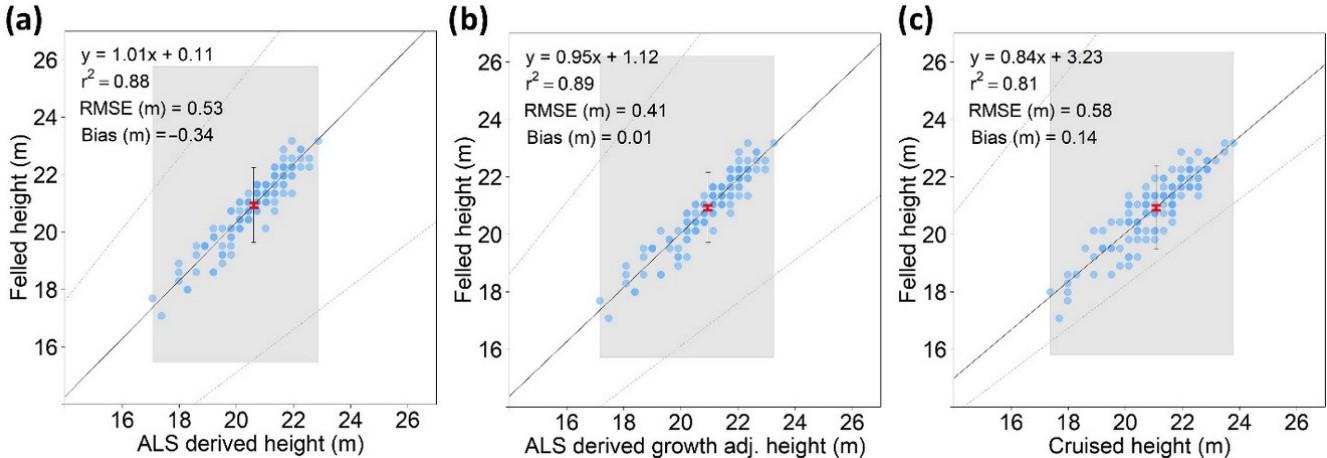

**Figure 4.** Equivalence test graphs for felled tree height versus (**a**) ALS-derived individual tree height, (**b**) ALS-derived growth-adjusted individual tree height and (**c**) manually measured "cruised" individual tree height. The grey polygon represents the $\pm 25\%$ region of equivalence for the intercept. The ALS-derived and cruised height are equivalent to the felled height when the vertical red bar is completely within the grey polygon. The grey dashed lines represent the $\pm 25\%$ region of equivalence for the slope. If the vertical black bar is within the grey dashed lines, then the regression slope is significantly similar to 1. The solid black line represents the best-fit linear regression model.

Figure 6 presents results from the regression-based equivalence tests for DBH. For both paired DBH datasets (ALS-derived vs. cruised and ALS growth-adjusted vs. cruised), the 95% confidence interval for the intercept (vertical red bar) was within the $\pm 25\%$ region of equivalence (grey polygon), indicating that the null hypothesis of dissimilarity can be rejected (i.e., paired datasets are equivalent). For both the ALS-derived non-growth-adjusted DBH (Figure 6a) and growth-adjusted DBH (Figure 6b), the 95% confidence interval for the slope (vertical black bar) was not completely within the $\pm 25\%$ region of equivalence (grey dashed lines), indicating slopes were not significantly similar to 1. The linear relationship between ALS-derived and cruised DBH (Figure 6a) and the relationship between ALS-derived growth-adjusted and cruised DBH (Figure 6b) both had an $R^2$ value of 0.3.

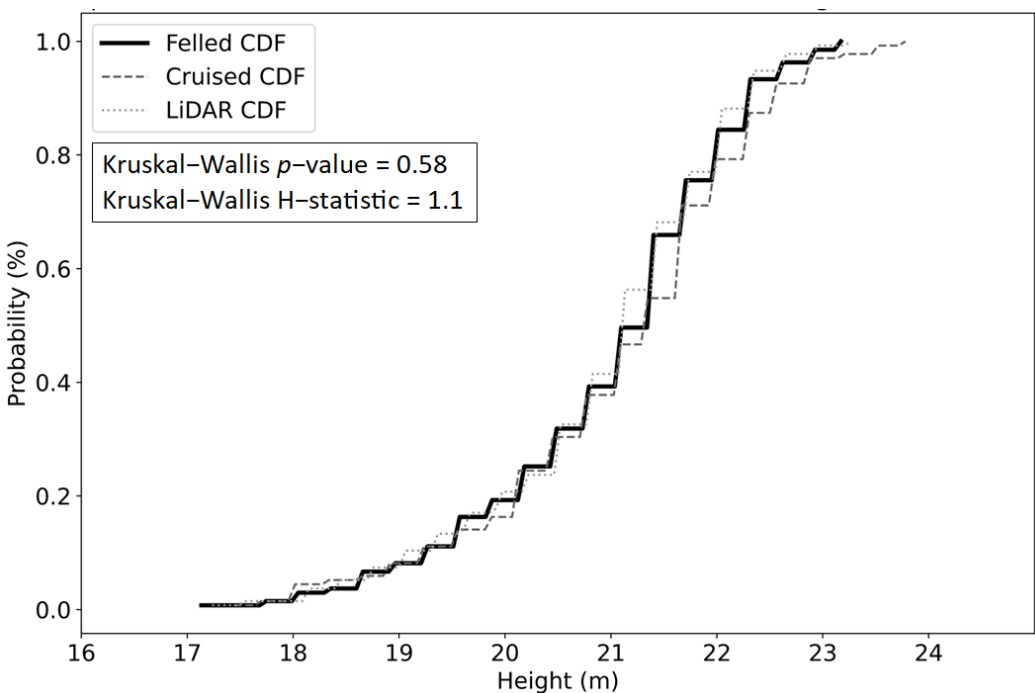

**Figure 5.** Empirical Cumulative Distribution Function (eCDF) for tree heights derived by ForestView® from LiDAR, cruise measurements, and felled tree measurements. Tree heights within each dataset appear to follow a standard normal distribution and are not significantly different (Kruskal-Wallis *p*-value: 0.58, H-statistic: 1.1).

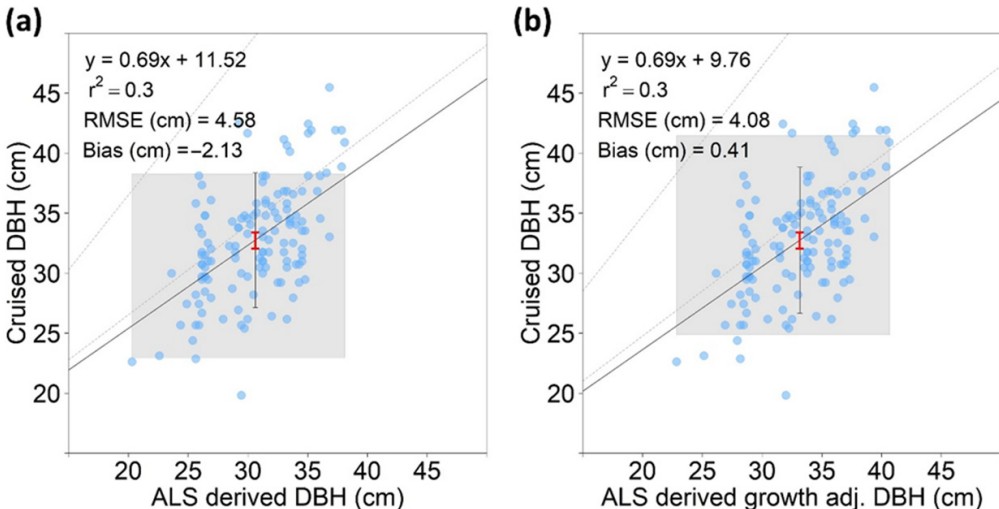

**Figure 6.** Equivalence test graphs for cruised tree DBH versus (**a**) ALS-derived DBH and (**b**) ALS-derived growth-adjusted DBH. The grey polygon represents the ±25% region of equivalence for the intercept. The ALS-derived DBH is equivalent to the cruised DBH when the vertical red bar is completely within the grey polygon. The grey dashed lines represent the ±25% region of equivalence for the slope. If the vertical black bar is within the grey dashed lines, then the regression slope is significantly similar to 1. The solid black line represents the best-fit linear regression model.

The ALS-derived DBH of detected trees exhibited a −2.13 cm bias and RMSE of 4.58 cm compared to the cruised tree measurement data (Table 1, Figure 6a). Because the cruised tree data were collected after the beginning of the growing season, FPS growth software [40] and the work by [12] were used to estimate a *P. taeda* and DBH increment increase for all ALS-detected trees. This resulted in an average 2.54 cm increase in diameter

and 65 cm increase in height across the 139 felled trees. The ALS-derived growth-adjusted DBH of detected trees exhibited a 0.41 cm bias and RMSE of 4.08 cm compared with the cruised tree measurement data (Table 1, Figure 6b).

A comparison of the growth-adjusted ALS and field-measured DBH values were shown to differ slightly along a distribution of size classes (Kolmogorov–Smirnov *p*-value: 0.182, D-statistic: 0.13) (Figure 7). Additionally, within the field data (Table 1) and Figure 7, a positive bias existed for ALS-measured trees exhibiting a <32 cm DBH and ≤21.5 m height, and negative bias was observed for ALS-measured trees >32 cm exhibiting a ≥20.2 m height.

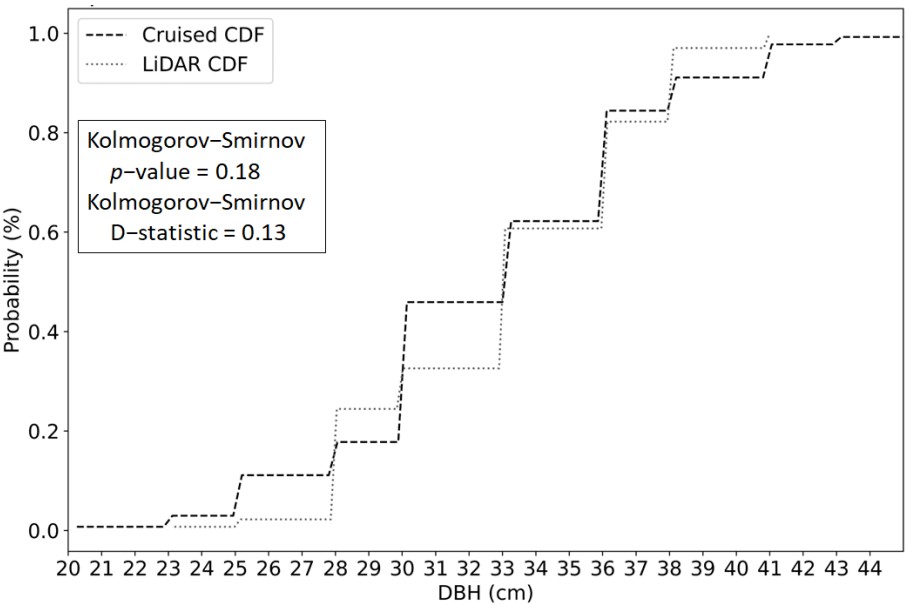

**Figure 7.** Empirical Cumulative Distribution Function (eCDF) for individual tree DBH values derived from ForestView® and cruised measurements. Tree DBH values within each dataset generally follow a standard normal distribution and were not significantly different within size classes (Kolmogorov-Smirnov *p*-value: 0.182, D-statistic: 0.13).

The ALS-derived volume of detected trees exhibited a 0.01 m$^3$ bias and RMSE of 0.15 m$^3$ compared to the felled tree measurement data. The cruised tree volume exhibited a 0.003 m$^3$ bias and RMSE of 0.03 m$^3$ compared with the felled tree measurement data. Figure 8 presents results from the regression-based equivalence tests for tree volume. For both paired volume datasets (ALS-derived vs. felled and cruised vs. felled), the 95% confidence interval for the intercept (vertical red bar) was within the ±25% region of equivalence (grey polygon), indicating that the null hypothesis of dissimilarity can be rejected (i.e., paired datasets are equivalent). For ALS-derived volume vs. felled volume (Figure 8a), the 95% confidence interval for the slope (vertical black bar) was not completely within the ±25% region of equivalence (grey dashed lines), indicating the regression slope was not significantly similar to 1. Conversely, for cruised volume vs. felled volume (Figure 8b), the 95% confidence interval for the slope (vertical black bar) was within the ±25% region of equivalence (grey dashed lines), indicating the regression slope was significantly similar to 1. The linear relationship between ALS-derived and felled tree volume had R$^2$ value of 0.4, while the relationship between cruised and felled tree volume had a R$^2$ value of 0.98.

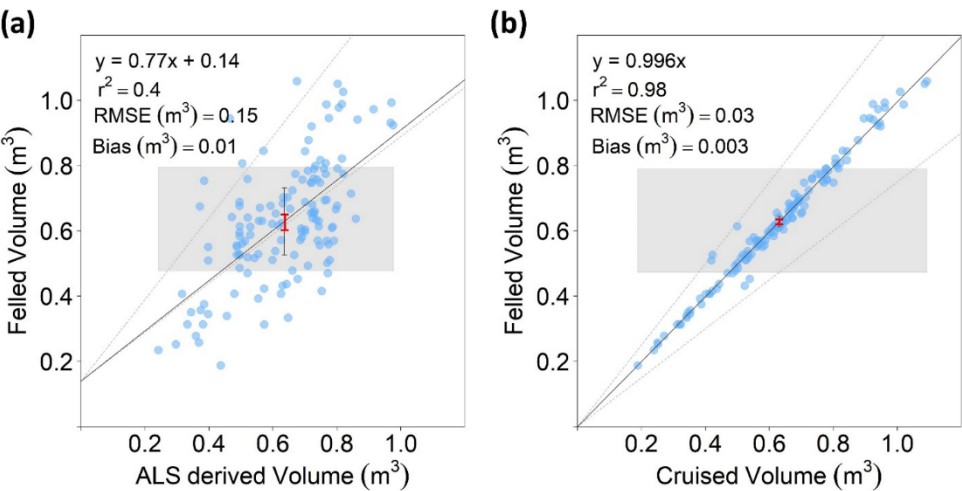

**Figure 8.** Equivalence test graphs for felled tree volume versus (**a**) ALS-derived volume and (**b**) cruised volume. The grey polygon represents the ±25% region of equivalence for the intercept. The ALS-derived and cruised volume are equivalent to the felled volume when the vertical red bar is completely within the grey polygon. The grey dashed lines represent the ±25% region of equivalence for the slope. If the vertical black bar is within the grey dashed lines, then the regression slope is significantly similar to 1. The solid black line represents the best-fit linear regression model.

## 4. Discussion

Detailed information is required across the operational forest industry sector to achieve production efficiencies [7,45] and inform investments in forest inventory type and scheduling [46]. Where data are limited, too costly, or of insufficient resolution, mangers experience added challenges that often lead to less effective decisions [5,47]. The increased application of wall-to-wall remote sensing data to provide forest inventory metrics has the potential to optimize timber harvest scheduling, forest investments, and the wood supply chain from forests to facilities [2,48]. To achieve efficiency gains throughout the supply chain, accurately measured forest inventory information such as tree height, DBH, and volume must be available to managers, and at sufficient spatial and temporal resolutions [48]. In this pilot study, we compared the results of an emerging commercial ITD software, ForestView®, against conventionally cruised and felled-tree data collections to evaluate the accuracy of this ITD software in providing individual-tree heights and DBHs for a stand of *P. taeda* in east Texas. The results were further assessed by inputting ForestView® and felled-tree-derived height and DBH values into FPS, modeling for gross volume and then comparing FPS outputs for both felled tree gross volume and ForestView® gross volume. FPS is most commonly used within the western U.S. and the models it applies are proprietarily protected from calibration to other geographies; thus, this is a potential source of error in the volume presented in this study. As individual tree allometric data can be derived from LiDAR ITD models, future research should focus on the development of volume estimation that uses population-level information in place of the sample-driven models currently applied within forest industry. The ALS used in this study was of comparatively greater pulse density (20 ppm) than many of the datasets used in previous studies that also used fixed-wing aircraft [3,22,34,49–51]. Ten-fold and greater pulse density (200 ppm) ALS are available using helicopter [52] and UAV [53] aerial survey platforms, respectively. However, greater sampling time and cost, coupled with smaller sampling extents, may preclude their application on many of the operational landscapes (≥10,000 ha) of commercial forestry in North America. In this study the application of UAV-acquired video was of significant value in matching felled trees with ForestView® detected trees. This study did not have access to an ALS unit for the UAV. On-site piloting, battery power, and weather played a significant role in the quality and cost of the data collected for this project. Additionally, individual tree locations within this dataset were field verified with a differential GPS and

contained positional errors ranging from ~0.01 m to ~0.12 m (with 131 of the locations being more accurate than 10 cm) depending on the canopy density, tree lean and other factors at the time of sampling. We acknowledge tree lean and canopy density may have introduced an unknown degree of error into our tree-matching methodology and further emphasized the value of the UAV imagery. We also acknowledge the use of high-precision field georeferencing of individual trees played a critical role in quantifying the "match" and subsequent error of ForestView® tree objects to field-measured trees. Quantification of georeferencing and matching errors will be needed as LiDAR-derived tree maps are scaled up to larger spatial scales than the pilot site in this study. The combination of high-resolution (20 ppm) LiDAR and ForestView® software shown substantial improvement in tree-to-tree alignment compared to previous studying of DBH from LiDAR in east Texas *P. taeda* where a <3 ppm scan and multispectral imagery enabled the alignment of only 28% of a 155-tree study [32].

In terms of individual tree height measurement, ALS-derived datasets have been shown to be less costly and more accurate than conventional forest inventory practices. Conventional forest inventory approaches are sample-based, time and cost intensive, and are commonly executed over the span of 5 to 10 years. Additionally, traditional field measurements of tree height generally apply a triangulation method that is labor-intensive and can result in errors of 1 m to 5 m depending on canopy structure and occlusion of treetops from the position of the observer [27,54,55]. Across the forest industry, height measurements derived from Vertex laser rangefinders, relaskops, and clinometers are broadly accepted within operational forest inventories. Field measurements of individual-tree specific attributes are also generally understood within both industry and academia to include human-related systematic error [29] and objective bias dependent on field personnel experience [56]. Individual tree height derived from ALS data has been found to be highly accurate (i.e., height RMSE typically less than 1.5 m) [8,17,25,29]. Furthermore, several studies have demonstrated that ALS-derived tree height can be more accurate, and have less bias, than field-measured tree height [24,27]. Likewise, the ForestView®-derived tree heights in this study had lower RMSE (0.40 m) than cruised measurements (RMSE = 0.57 m) when compared with felled tree data, and aligned with the acceptable levels of accuracy expected by the commercially operational forest sector in the region where this study was conducted.

The equivalence tests indicated that mean ALS-derived DBH and ALS-derived growth-adjusted DBH were equivalent to field-measured DBH; however, regression slopes were not equivalent. This finding indicates that these models likely underpredict higher observations and overpredict lower observations [44]. This result was not unexpected given the temporal difference between the ALS acquisition and field data collection periods. This finding also highlights the importance of timing field data collection with remote sensing acquisitions to reduce uncertainty, which aligns with other studies [29,57,58]. Tree height and DBH are highly correlated [27,31,59], and many models utilizing remotely sensed data for DBH rely on accurate tree, canopy, or full stand-level heights to have confidence in predicted DBH [31,60–62]. Given the growth rates of *P. taeda* within the Nacogdoches, TX area [12], and likely throughout much of the industrial forestlands of the southeastern USA, our decision to nominally increase the DBH of each tree for the purposes of comparing volume is justified, but is a potential source of error. Furthermore, there is likely error associated with the DBH models that ForestView® applies, but we were not able to quantify this error given the proprietary nature of this software. The results showing equivalence between mean ALS-derived DBH and cruised DBH and ALS-derived growth-adjusted DBH are, however, promising for volume calculation as well as highlighting the potential errors associated with using remotely sensed and ground-based data with temporal discrepancies.

Tree height and DBH are inextricably related to stem volume and operational feasibility within a commercial forest. The results of the equivalence tests indicate that both ALS-derived volume and cruised volume are equivalent to felled tree volume. The linearity of FPS modeled volumes can be partially explained by the work of others [29,31,34,50], where the use of certain regression techniques within an architecture designed for sample

data, and inputs of average descriptors for aggregate species, density, diameter, or height classifications can result in predictable trends. The variability between gross tree volume from felled tree measurements and those of ForestView® is likely resulting from the natural DBH variation in trees of similar height and variation in ALS pulse density, which can impact error associated with inputs to allometric equations (e.g., tree height) [1,16,55,63]. Additionally, the taper and volume equations applied within the FPS model were broadly defined for *P. taeda* within the southeastern U.S. and could be improved with regional or local calibration data. The volume comparisons performed in this study are intended to help further our understanding of sample- versus population-modeling given the increased use of remote sensing in forestry [2,48,64,65] and biomass/carbon applications [3,32,66]. Additional research focused on stem volume modeling within FPS would be required to further characterize variance observed in this study.

The height, DBH and volume results of this study and others using high-density ALS data indicate that ALS-derived individual tree segmentation can enable accurate derivation of essential individual-tree metrics over coniferous forests [5,21,48,49] within conditions similar to those of each respective study. Although we acknowledge that 139 felled trees may seem like a low sample size, felling this many mature trees for LiDAR ground truth assessments represents a considerable amount of effort that is rarely conducted. For example, an assessment in the northwestern United States used only a total of 60 trees, where only 24 were felled by the principal study and a further 36 were felled by a companion group [27]. A study in the Italian Alps conducted an accuracy assessment of LiDAR datasets using 100 felled trees [25], a study in Germany felled 30 trees to cross-compare the utility of LiDAR and UAVs to assess tree heights [29], and a study in Cameroon felled 61 trees [26]. As a result, most lidar accuracy assessment studies have focused on indirect field height measurements using laser rangefinders that can exhibit considerable uncertainties, especially when overtopped and suppressed trees are present, or the treetops are occluded by branches or other trees [24].

The alignment of multiple datasets (ALS, cruising, and felled-tree) within this study is not common across the remote sensing literature, and although the sample size is small compared to operational landscapes the findings align with the work of others such as [8] who identified strong correlations between individual tree attributes (height, volume) derived from individual tree segmentation and attributes derived from conventional forest inventory across a large-scale forested area (24,220 ha). The results from our study and the review of others suggests individual-tree processing of ALS data can offer a cost-effective alternative to conventional forest inventories. Throughout the world, the application of ALS data processing in support of commercial forest inventories has grown since the early 2000s and now supports a number of stand-based inventory examples in many countries by leveraging area-based imputation methods [5,48]. As the availability of high-resolution ALS data increases and acquisition costs decrease, accuracy of individual-tree metrics (e.g., height, DBH, volume), is anticipated to improve. These improvements will likely lead to an increase in operational ALS-based ITD routines on commercial forests in place of conventional sample-based inventories. Lastly, as ALS-derived maps of individual trees and their corresponding attributes become available, forest managers and landowners will gain access to information far beyond what conventional forest inventories can provide. This may enable managers to evaluate disease, insect, fire, and forest management impacts on individual tree productivity and mortality at landscape spatial scales and lead to improved forest modeling and management.

## 5. Conclusions

The objectives of this pilot study were to assess the accuracy of ForestView®-derived individual tree attributes, including height, DBH and volume, for a *P. taeda* stand in eastern Texas. This research shows that height, DBH and volume derived using ForestView® are equivalent to both conventional cruised and felled tree measurements. The resulting ALS-derived heights from this study were more accurate than heights collected through

conventional cruise sampling, and highlight the potential of using ForestView® for measuring individual tree attributes in similar pine stands where high-resolution LiDAR is available [29]. While ForestView® measurements were equivalent to felled tree measurements, observed positive and negative DBH value bias from the ForestView® model at each end of the tree-height distribution suggests the model may not have fully captured the natural variation of DBH within this small *P. taeda* test area. Future research should apply ForestView® at landscape (>5000 ha) scales to confirm these observations and provide additional insights into the scalability of this software as well as test the accuracy of tree counts as identified by the software against on-ground counts without the aid of a 100% geolocated field population. Additionally, given the need for accurate height and DBH measurements for calculating tree volume, future research should consider the use of ALS-derived individual tree-based methods to investigate gross tree volume produced by traditional forest models such as FPS. Future work should also consider a comparative analysis of ITD methods, leveraging high-resolution LiDAR, to traditional stand-based inventory methods at a landscape scale to better differentiate population- and sample-based forestry approaches and further assess the scalability of these tools. This comparative analysis would provide insights to managers and conservationists seeking a greater resolution of forest heterogeneity to better quantify volume, biomass, carbon, and ultimately value, within forested settings.

**Supplementary Materials:** The following supporting information can be downloaded at: https://www.mdpi.com/article/10.3390/rs14112567/s1, Table S1: Felled, cruised, and Lidar-derived tree measurement data.

**Author Contributions:** Conceptualization M.V.C.; methodology M.V.C. and A.M.S.; software M.V.C., A.M.S. and A.M.S.S.; validation M.V.C. and A.M.S.; formal analysis A.M.S. and M.V.C.; investigation, data resources and data curation M.V.C. and A.M.S.; writing—original draft preparation M.V.C.; writing—review and editing A.M.S.S. and A.M.S.; visualization A.M.S.; supervision and project administration M.V.C.; funding acquisition None. All authors have read and agreed to the published version of the manuscript.

**Funding:** This research received no external funding.

**Data Availability Statement:** The tree measurement data presented in this study are available in Supplementary Table S1.

**Acknowledgments:** We would like to thank biometrician Albert Pancoast for his review and comments on tree allometry and volume modeling results throughout the development of this analysis.

**Conflicts of Interest:** The authors declare no conflict of interest.

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
