# Peer review of "A Conventional Cruise and Felled-Tree Validation of Individual Tree Diameter, Height and Volume Derived from Airborne Laser Scanning Data of a Loblolly Pine (P. taeda) Stand in Eastern Texas"

_remotesensing, doi:10.3390/rs14112567_

Round 1
Reviewer 1 Report
the article is interesting and presents robust data. I recommend better describing the methodology used in ForestView. Improve discussion of results with more citations. Add an updated discussion from 2021-2022 to the introduction.Author Response
- The article is interesting and presents robust data. I recommend better describing the methodology used in ForestView.
Author Response (AR) 1: We have re-written the ALS methods section of the manuscript (Section 2.3), where ForestView is described, to improve clarity/relevance to readers.
Reviewer 2 Report
Correct text fragments that come from the MDPI template.
Correct the sections.
On lines 111-112 it says: "The measurement stand was comprised of six rows planted at a 3-meter square spacing." Is this correct or is the density too high?
Correct numerous reference errors.
Author Response
- Correct text fragments that come from the MDPI template.
Author Response (AR) 1: Completed.
- Correct the sections.
AR2: We have corrected the section headings. The section “methods” has been changed to “materials and methods” with sub headers for “Study area”, “ALS data”, ALS Single-Tree detection and Measurement”, “Field validation data”, and “Height, DBH, and Volume accuracy assessments”.
- On lines 111-112 it says: "The measurement stand was comprised of six rows planted at a 3-meter square spacing." Is this correct or is the density too high?
AR3: We have revised the tree spacing value after discussion with the private landowner. They confirmed the trees in this stand were planted at a 12-foot square spacing and as such we have revised the text to reflect a 3.7m value. Thank you for helping us clarify this measurement.
- Correct numerous reference errors.
AR4: Corrected. We have fixed the references in the revised manuscript and have included a .PDF with our resubmission to ensure reviewers have a formatted and readable version
Reviewer 3 Report
Comment
This manuscript focused on conventional cruise and felled-tree validation of individual tree derived from airborne laser scanning data for a Loblolly pine stand. However, I fell that the sample size is too small and scientific sound is weakness in remote sensing fields. I reviewed it and provided some special comments as follows.
- Although Title and Abstract could reflect whole text but Title is not so sound in remote sensing fields.
- I could understand ForestView®; therefore, I don’t understand why the study purpose is important.
- Materials and Methods (Line 91) should move to next line.
- Repeated Materials and Methods was shown in Lines 91 and 108.
- The 3 data type, e.g., felled, ALS derived and cruised volume, was shown in Materials and Methods chapter. However, those data were not surveyed at same time? Lines 170-171 “Lastly, given the temporal discrepancy between the ALS data collection (December 2020) and the felled tree data (May 2021)”.
- I suggested that authors should describe above 3 data type more detailed in Materials and Methods chapter.
- The sample size (139 trees) is too small in remote sensing study.
- Too many “Error! Reference source not found” were shown in text. It should be improved!
- Figure 3 with 3 colors but the legend only notes 2 items in this Figure?
Overall, I review this manuscript but the sample size is too small and scientific sound is weakness. Therefore, it is not suitable for publication in the remote sensing.
Author Response
This manuscript focused on conventional cruise and felled-tree validation of individual tree derived from airborne laser scanning data for a Loblolly pine stand. However, I fell that the sample size is too small and scientific sound is weakness in remote sensing fields. I reviewed it and provided some special comments as follows.
- Although Title and Abstract could reflect whole text but Title is not so sound in remote sensing fields.
Author Response (AR) 1:
After much thought on a revised title, we have agreed as authors to maintain the title as submitted because of two details, 1) the variations in tree attributes (DBH and height) observed on the ground align with the true variability in single-tree diameter occurring throughout commercial Loblolly plantations of the southeastern US based on our experience and that of others in the published literature, specifically the work of Liu, J.P., Burkhart, H.E., 1994. Spatial autocorrelation of diameter and height increment predictions from 2 stand simulators for loblolly pine. For. Sci. 40, 349. And 2) because the felling and measurement of 139 mature commercial plantation pines in combination with a 100% cruise measurement pre-felling, and the occurrence of a 20ppm aerial Lidar dataset from a fixed wing aircraft within the nearly temporal synchronization of this study was not found to have an equal throughout the literature review we performed, globally.
- I could understand ForestView®; therefore, I don’t understand why the study purpose is important.
AR2: We have re-written the ALS methods section of the manuscript (Section 2.3), where ForestView is described, to improve clarity/relevance to readers. We hope the revisions and additions throughout the revised manuscript help to clarify the relevance and purpose of this work and thank the reviewers for prompting some additional efforts in this regard.
- Materials and Methods (Line 91) should move to next line.
AR3: Corrected.
- Repeated Materials and Methods was shown in Lines 91 and 108.
AR4: Corrected.
- The 3 data type, e.g., felled, ALS derived and cruised volume, was shown in Materials and Methods chapter. However, those data were not surveyed at same time? Lines 170-171 “Lastly, given the temporal discrepancy between the ALS data collection (December 2020) and the felled tree data (May 2021)”.
6.I suggested that authors should describe above 3 data type more detailed in Materials and Methods chapter.
AR5-6: We have substantially redeveloped the methods sections of this manuscript to more clearly define research efforts, timelines, and details of methodologies. As part of this, we have explicitly stated when each dataset was collected in the field validation dataset section (Section 2.4).
- The sample size (139 trees) is too small in remote sensing study.
AR7: As noted by the Editor, felling trees is a considerable effort that is rarely conducted, with most studies relying on laser rangefinders to provide indirect measures that can exhibit considerable uncertainties (i.e., 10-20% in total height, especially if trees are occluded by branches and other trees). Although we found one study that planted 450 seedlings and scanned them after 2 years, most of the studies assessing mature trees felled under 100 trees, making this study at the high end for sample size.
We however agree with the reviewer’s sentiment that in an ideal world, all lidar validation studies should fell trees. We would welcome the journal requiring such a standard as we believe it would help place the current study in context given felling 139 trees represents a high sample size rarely conducted in the remote sensing literature to validate lidar.
To address this concern we have added the following text to the methods:
Although we acknowledge that 139 felled trees may seem like a low sample size, felling this many mature trees for LiDAR ground truth assessments represents a considerable amount of effort that is rarely conducted. For example, an assessment in the northwestern United States used only a total of 60 trees, where only 24 were felled by the principal study and a further 36 were felled by a companion group [Tinkham et al. 2016]. A study in the Italian Alps conducted an accuracy assessment of LiDAR datasets using 100 felled trees [Sibona et al. 2016], a study in Germany felled 30 trees to cross-compare the utility of LiDAR and UAVs to assess tree heights [Ganz et al. 2019], and a study in Cameroon felled 61 trees [Takoudjou et al. (2017]. As a result, most lidar accuracy assessment studies have focused on indirect field height measurements using laser rangefinders that can exhibit considerable uncertainties, especially when overtopped and suppressed trees are present or the tree tops are occluded by branches or other trees (Wang et al. 2019).
- Too many “Error! Reference source not found” were shown in text. It should be improved!
AR8: Corrected. We have fixed the references in the revised manuscript and have included a .PDF with our resubmission to ensure reviewers have a formatted and readable version.
- Figure 3 with 3 colors but the legend only notes 2 items in this Figure?
AR9: Figure 3 was built with a “transparency” on the LiDAR data overlain on the Felled data, thus where they overlap a different color is visualized. We have added another row to the legend of this figure for clarity. Thank you for noting this misconception.
Round 2
Reviewer 3 Report
This manuscript was improved following my suggestion. My main concern is still on that the sample size is too small. Nevertheless, authors could point out it in Discussion or Conclusions chapter. On the other hand, new finding of this manuscript could be emphasized in Conclusions.
Author Response
Dear Reviewer,
We have added a clarifying section to the discussion of the manuscript from lines 458 to 476 that expand on the sample size of this work and comparison to previous studies. Thank you kindly for the feedback and comments that improve the readability of this work.
Sincerely,
The Authors.
